# The Impact of Late Follicular Phase Progesterone Elevation on Cumulative Live Birth Rate and Embryo Quality in 4072 Freeze-All Cycles

**DOI:** 10.3390/jcm11247300

**Published:** 2022-12-08

**Authors:** Ling Huang, Sunxing Huang, Yangxing Wen, Xiubing Zhang, Xiaokun Hu, Rihan Wu, Minghui Chen, Canquan Zhou

**Affiliations:** 1Reproductive Medicine Center, First Affiliated Hospital, Sun Yat-Sen University, No. 58 Zhongshan Road 2, Guangzhou 510080, China; 2Department of Reproductive Immunity, Guangdong Women and Children Hospital, No. 521 Xingnan Road, Guangzhou 511400, China

**Keywords:** progesterone level, ovarian stimulation, embryo quality, cumulative live birth rate, freeze-all strategy

## Abstract

Late follicular phase progesterone elevation during in vitro fertilization impedes embryo implantation. It is unclear whether late follicular phase progesterone elevation still has a negative effect on cumulative live births and embryo quality when a freeze-all strategy is adopted. Data from a total of 4072 patients were reviewed. All patients used the freeze-all strategy. Multivariate regression analyses were used to assess the association of progesterone levels with both cumulative live birth and embryo quality. There was no significant difference in the cumulative live birth rate between the groups with progesterone level <1.5 ng/mL and ≥1.5 ng/mL. The progesterone level was not associated with cumulative live birth and embryo quality.

## 1. Introduction

Late follicular phase progesterone elevation (LFPE) is commonly defined as having a progesterone (P) level ≥ 1.5 ng/mL (4.77 nmol/L) on the day of human chorionic gonadotropin (HCG) administration, and its incidence rate has been reported to be 6–30% in controlled ovarian stimulation [1]. It has been reported that the number of preovulatory follicles is significantly associated with the concentration of circulating progesterone, indicating that each individual follicle contributes to the circulating levels, and the more follicles there are, the higher the progesterone output will be [2,3]. The impact of LFPE on the clinical outcome of in vitro fertilization (IVF) cycles has been reported by many studies, yielding conflicting results [4,5,6].

One of the proposed impacts of LFPE is its effect on endometrial receptivity. The previous studies demonstrated a distinct difference in endometrial gene expression profiles between patients with a progesterone serum concentration above and below the threshold of 1.5 ng/mL on the trigger day and LFPE may alter endometrial receptivity [7,8].

The impact of LFPE on the quality of the oocyte and embryo has been another hot issue of concern. Previous studies have used an oocyte donation model to assess the influence of elevated progesterone on cycle outcome. A retrospective analysis that included 397 cycles for fresh oocyte donation showed that a P level > 1.5 ng/mL did not have an influence on embryo quality or cumulative live birth rate (CLBR) in recipient cycles [4]. However, the study conducted by Yovel et al. holds the opposite opinion that exposure to elevated progesterone had detrimental effects on oocytes [9].

Although previous studies have reported the live birth rate of patients with late follicular phase progesterone elevation [10,11], the evidence showing its impact on frozen embryo transfer (FET) cycles is quite limited [12]. Since progesterone elevation negatively affects endometrial receptivity, the freeze-all strategy is the usual alternative [13,14]. Therefore, the cumulative live birth rate (giving complete information about the reproductive outcomes out of a whole controlled ovarian stimulation cycle) is of great need. However, there are few investigations on the cumulative live birth rates of patients with LFPE in freeze-all cycles.

As the freeze-all strategy was carried out when the progesterone level on the trigger day was 1.5 ng/mL or higher in our institute, the primary purpose of our study was to investigate the association between P level on trigger day and cumulative live birth rate. The secondary aim of our study was to assess the influence of P level on the trigger day on embryo quality.

## 2. Materials and Methods

### 2.1. Patients Population

This was a retrospective, single-center study. Our patient population consisted of women who took part in the freeze-all strategy and underwent their first IVF or ICSI using the long gonadotropin-releasing hormone (GnRH) agonist or GnRH antagonist protocol between January 2013 and December 2017 at the Reproductive Medicine Center of the First Affiliated Hospital of Sun Yat-sen University. Cycles with testicular sperm extraction or preimplantation genetic testing, or involving oocyte donation, oocyte sharing, oocyte cryopreservation, and/or frozen oocyte thawing were excluded from the analysis. The patients were categorized into two groups based on their trigger day P levels: the <1.50 ng/mL group and the ≥1.50 ng/mL group. A cutoff of 1.5 ng/mL was used according to the results of previous studies reporting poorer cycle outcomes with higher P levels on the day of the HCG trigger [10,15].

### 2.2. Hormone Assays

On the morning of the trigger day, the serum progesterone concentration of each patient was determined using a chemiluminescent microparticle immunoassay (Architect System, Abbott Laboratories, Chicago, IL, USA) with a sensitivity of 0.005 ng/mL. The intra- and interassay coefficients of variation were 1.3% and 3.9%, respectively. There was no change in the assay during the study, and the laboratory regularly performed a calibration to minimize variations in the results associated with time and reagent batch renewal.

### 2.3. Controlled Ovarian Stimulation Protocol

Ovarian stimulation was performed using a long GnRH agonist or GnRH antagonist protocol. In the long GnRH agonist protocol, a long-lasting formulation of triptorelin acetate depot (1.0–1.8 mg) or a daily dose (0.05–0.1 mg) of triptorelin acetate was used for pituitary downregulation, and gonadotropin stimulation was initiated 14 days later. In the GnRH antagonist protocol, gonadotropin stimulation was initiated on cycle day 2. Subcutaneous GnRH antagonist was started on the 5th day of gonadotropin stimulation with a daily dose of 250 μg. The dose of recombinant FSH (150–300 IU) was determined based on the patient’s age, weight, and ovarian reserve, with or without human menopausal gonadotropin. Final oocyte maturation was induced by administering HCG (250 ug) when at least two follicle ≥ 18 mm in diameter or three follicles ≥ 17 mm in diameter could be visualized on ultrasonography. Oocyte retrieval was performed 34–36 h after HCG administration. Fertilization was performed using either standard insemination or ICSI.

### 2.4. Freeze-All Strategy

If the P level was 1.5 ng/mL or higher on the trigger day, fresh embryo transfer was cancelled and all embryos were frozen. Physicians also cancel fresh embryo transfer and freeze all the embryos according to the risk of ovarian hyperstimulation syndrome (OHSS), the presence of irregular endometrium such as the endometrial fluid on the day of oocyte retrieval, the presence of endometrial polyp and sub-mucosal myomas, the physical condition of patients, clinical preference, and other nonspecific factors.

### 2.5. Vitrification and Preparation of the Frozen Embryo Transfer Cycle

Among Day 3 embryos graded by Cummins’s criteria [16], two top-quality embryos were selected for vitrification, while the remaining embryos were subjected to extended culture. Morphologically good blastocysts (grade ≥ 3 BC) according to the Gardner scoring system [17] were then vitrified on Day 5 or 6. Frozen embryo transfer protocols included the natural cycle and the hormone replacement therapy cycle. For the natural cycle regimen, ovulation was determined by ultrasound monitoring and hormonal assessment of the spontaneous luteinizing hormone (LH) surge and P elevation. Luteal phase support was started after the ovulation day with oral dydrogesterone 10 mg twice a day. For the hormone replacement therapy cycle regimen, the endometrium prepared with oral estradiol valerate at a dose of 4–9 mg daily was started on day 2–5 of the menstrual cycle. Intramuscular injections of progesterone (40 mg/day) were added when the endometrial thickness reached 7 mm or more, and two days later the dose was increased to 60 mg daily. Embryo transfer was performed on the 3rd or 5th day of ovulation, and on the 4th or 6th day of progesterone initiation. The single blastocyst was transferred according to the score from best to worst. Two cleavage embryos were transferred finally when all blastocyts had been transferred without a live birth.

### 2.6. Definition of High-Quality Embryos

Embryo morphology was assessed on day 3, day 5, and day 6. Cleavage-stage embryos with at least 7 blastomeres and fragmentation < 20% were defined as high-quality embryos [16]. Blastocysts were scored based on the Gardner and Schoolcraft grading system [17] and regarded as high quality if they reached at least an expansion stage 4 with A or B for inner cell mass and trophectoderm.

### 2.7. Outcome Measures

The primary outcome measure was CLBR per ovarian stimulation cycle. The secondary outcome was the number of viable embryos and high-quality embryos. No cumulative live birth means no birth at all in one ovarian simulation cycle. A live birth was defined as a birth event in which at least one baby was born alive. We determined CLBRs by considering the number of first live births generated during the IVF/ICSI cycles as the numerator and the total number of ovarian stimulation cycles as the denominator; additional live births were not considered.

### 2.8. Statistical Analysis

The Kolmogorov–Smirnov test was applied to analyze the distribution of data in two-group comparisons. Normally distributed data are presented as the mean ± standard deviation of the mean and compared with a two-tailed Student’s *t*-test. Skewed data are presented as the mean and quartiles and examined by Mann–Whitney U tests. Categorical data were presented as frequencies and percentages within each study group. Intergroup differences were assessed using chi-squared tests for categorical data. Fisher’s exact test was used when the expected values in any of the cells of a contingency table were below 5; in all other cases, Pearson’s chi-squared test was applied. Multivariate logistic regression analysis was conducted to explore the association between P level on the trigger day and the cumulative live birth rate. Multivariate linear regression analysis was used to assess the association between P levels on the trigger day and the number of viable and high-quality embryos. Variables which were significant in the univariate logistic regression analysis and clinically relevant to CLBR were encompassed as confounding factors in the multivariate logistic regression models. The confounders included in the multivariate linear regression model were female patient age and number of retrieved oocytes which have been consistently shown to be associated with embryo quality [18,19]. Statistical significance was set at *p* ≤ 0.05. Analyses were performed using IBM SPSS statistics (version 25; IBM, Chicago, IL, USA).

## 3. Results

### 3.1. Study Population

A total of 4072 patients who took part in the freeze-all strategy and underwent IVF or ICSI using the long GnRH agonist or GnRH antagonist protocol between January 2013 and December 2017 were included in the present study. A study flowchart was shown in Figure 1. Of the cycles included, the incidence of LFPE was 10.29%. If the P level was 1.5 ng/mL or higher on the trigger day, fresh embryo transfer was canceled and all embryos were frozen. The main indications to freeze-all in the control group include the risk of ovarian hyperstimulation syndrome (40.4%), irregular endometrium (25.4%), clinical preference (17.5%), physical condition of patients (5.9%), and others (10.8%).

#### 3.1.1. Baseline Characteristics and Outcome in Groups with P level < 1.5 ng/mL and ≥1.5 ng/mL on the Trigger Day

The patient’s age, the proportion of long GnRH agonist protocol, duration of stimulation, the total dose of gonadotropins administered, estrogen level on the day of HCG administration, number of oocytes retrieved, number of oocytes fertilized, and number of viable embryos and high-quality embryos in the P < 1.5 ng/mL group were significantly lower than those in the P level ≥ 1.5 ng/mL group. The LH level on the trigger day and body mass index in the P level < 1.5 ng/mL group were significantly higher than those in the P level ≥ 1.5 ng/mL group. The antral follicle count and basal FSH which are the markers of ovarian reserve are similar between the two groups. There was no significant difference in the CLBR between the two groups (Table 1).

#### 3.1.2. Association between P level on the Trigger Day and Cumulative Live Birth Rate

Univariate logistic regression analysis showed a significant association between CLBR and female age, BMI, stimulation protocol, total dose of gonadotropins, LH level on the trigger day, and the number of oocytes retrieved (*p* < 0.001). The multivariate logistic regression model, taking into consideration the above-mentioned confounders, demonstrated that the progesterone level on the day of HCG administration was not an independent predictive factor of CLBR. The adjusted odds ratio (OR) for the P level on the trigger day was 1.109 (0.988–1.245, *p* = 0.080) after adjustment for potential confounders (Table 2).

#### 3.1.3. Association between P level on the Trigger Day and Embryo Quality

Multivariable linear regression was conducted to explore the association between P level on the trigger day (P < 1.5 ng/mL or ≥1.5 ng/mL) and embryo quality. The linear regression was analyzed separately according to different ovarian stimulation protocols. After adjustment for potential confounders, P level (P < 1.5 ng/mL or ≥1.5 ng/mL) was not associated with the number of viable embryos. The P level (P < 1.5 ng/mL or ≥1.5 ng/mL) was not associated with the number of high-quality embryos after the adjustment for the female patient’s age and the number of retrieved oocytes (Table 3). The number of oocyte retrieved was positively associated with the number of viable embryos and high-quality embryos.

## 4. Discussion

In the present study, we used a freeze-all strategy in cases where trigger day P levels were ≥1.50 ng/mL and analyzed the influence of LFPE on cumulative live birth and embryo quality. Our results showed that the CLBRs were similar for the groups with trigger day P levels < 1.50 ng/mL and ≥1.50 ng/mL. The P level on the trigger day was not associated with the cumulative live birth or the number of high-quality embryos after adjusting for confounding factors.

The results of two previous retrospective studies showed that elevated serum progesterone on the trigger day adversely affected CLBs in autologous IVF/ICSI cycles with fresh embryo transfer [5,6]. In the present study, we used a freeze-all strategy if the trigger day P level was ≥1.50 ng/mL. Although there were statistically significant differences in the number of oocytes retrieved, fertilized, viable embryos and high-quality embryos, and the cumulative live birth rate of the groups with the trigger day P levels < 1.50 ng/mL and ≥1.50 ng/mL showed no statistical difference. As the baseline information between the two groups was different, we conducted multivariate logistic regression analysis to adjust for confounding factors. The result demonstrated that the P level on the trigger day was not an independent predictive factor of CLBR. The confounders included in the multivariate logistic model have been consistently reported to be associated with CLBR in previous studies [20,21,22]. The results of the present study are supported by another retrospective analysis of fresh oocyte donation cycles, which showed that excluding the detrimental effect of LFPE on endometrial receptivity, LFPE does not influence CLBR [4].

The impact of LFPE on embryo quality remains unconfirmed. Some previous studies had demonstrated a deleterious impact of LFPE on embryonic development potential and quality. Yovel et al. reported that the exposure to LFEP (threshold 1.9 ng/mL) was significantly associated with the detrimental effect on oocyte/embryo quality [9], but they did not conduct multivariate regression analysis to eliminate the interference from the confounding factors. Similarly, the study of Vanni et al. reported that progesterone levels at the induction showed an inverse relation with top quality blastocyst formation and ROC curve analysis identified P level > 1.49 ng/mL as the best cut-off for identification of patients at risk for the absence of top quality blastocysts [23]. However, only patients who underwent blastocyst culture were included in the study which might cause selection bias. Huang et al. retrospectively analyzed 4236 fresh IVF cycles. Although the results showed that the proportion of top-quality embryos was negatively associated with LFPE, the confounder included in multivariate logistic model of the study was only the duration of infertility [24]. Nevertheless, some studies held the opposite opinion. Hernandez-Nietoetal retrospectively analyzed 5244 controlled ovarian hyperstimulation cycles for IVF/preimplantation genetic testing for aneuploidy, and after adjusting for age, BMI, anti-müllerian hormone, baseline FSH, days of stimulation, and oocytes retrieved per case, there was no association found between LFPE and increased odds of embryo aneuploidy. Their analysis also suggested that LFPE was not associated with impaired embryonic development, although they did not adjust for confounding factors [25]. In the present study involving 4072 freeze-all cycles, after adjusting for confounding factors, the multivariate linear regression analysis demonstrated that the P level on the trigger day was not associated with the number of viable and high-quality embryos in both GnRH agonist and antagonist protocol. The confounders including female patient age and number of retrieved oocytes have been consistently reported to be associated with embryo quality in the previous studies [18,19].

One of the strengths of the present study is its relatively large sample size. Additionally, in the present study, a freeze-all strategy was used when the P level on the trigger day was 1.5 ng/mL or higher, which has been found to negate the impact of LFPE on endometrial receptivity in the fresh embryo transfer cycle [14,26]. Another strength of this study is the use of multivariate logistic regression and linear regression analyses, which aim to statistically eliminate the interference from confounders or other forms of bias, to assess the true effect of elevated P levels on the day of HCG trigger on CLBR and embryo quality.

The limitations of the present study lie mainly in its retrospective design, which may cause bias. There are many other suggestions about the cut-off level of progesterone for deciding freeze-all strategy with no fresh embryo transfer [13], further studies with a different cut-off level of progesterone would be desirable to confirm our conclusions. Another limitation is that the definition of CLBR used here may underestimate the total live births during the ovarian stimulation cycle. However, the definition of CLBR in our manuscript was used widely in previous studies [27,28]. Additionally, the different choices of endometrial preparation and luteal phase support for frozen embryo transfer were not analyzed. The results of the meta-analysis by Ghobara, T et al., demonstrated that there was no evidence of the difference in the live birth rate between the natural cycle and artificial endometrial preparations [29]. Therefore, the cumulative live birth rate analysis may not be interfered with by different endometrial preparations.

## 5. Conclusions

In conclusion, there was no association found between LFPE and cumulative live birth when a freeze-all strategy was adopted in patients with LFPE during IVF/ICSI. Moreover, there was also no correlation found between LFPE and embryo quality in our study. Further studies with randomized clinical trials would be desirable to confirm our findings.

## Figures and Tables

**Figure 1 jcm-11-07300-f001:**
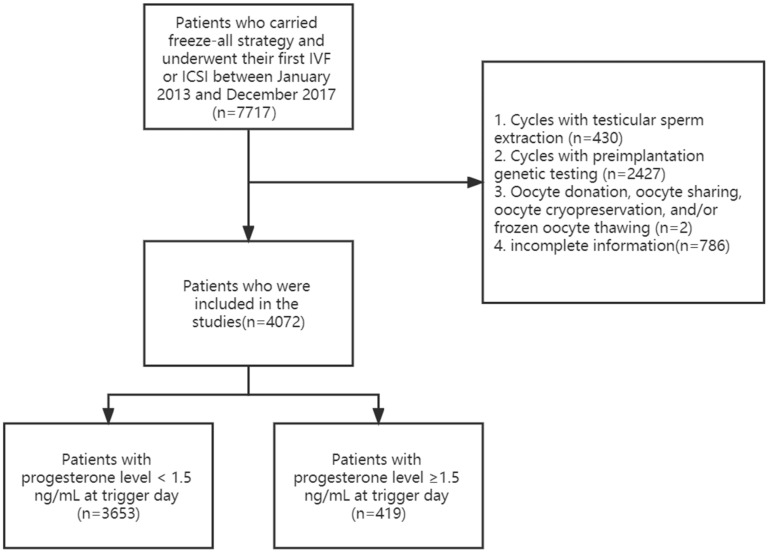
Flowchart showing the procedures used in selecting the studied patients.

**Table 1 jcm-11-07300-t001:** Patient characteristics and cumulative live birth rate in groups with progesterone level <1.5 ng/mL and ≥1.5 ng/mL at trigger day.

	Progesterone Level < 1.5 ng/mL (*n* = 3653)	Progesterone Level ≥ 1.5 ng/mL (*n* = 419)	*p* Value
Female age (years)	32 (28–35)	32 (30–36)	0.001 ^a,^*
Duration of infertility (years)	3 (2–6)	4 (2–6)	0.187 ^a^
Body mass index (kg/m^2^)	20.90 (19.40–23.00)	20.40 (19.10–22.31)	<0.001 ^a,^*
Type of infertility			
Primary	1756 (48.1%)	191 (45.6%)	0.335 ^b^
Secondary	1897 (51.9%)	228 (54.4%)	
Gravidity	1 (0–2)	1 (0–2)	0.525 ^a^
Parity	0 (0–0)	0 (0–0)	0.637 ^a^
Times of previous miscarriage	0 (0–1)	0 (0–1)	0.831 ^a^
Insemination method			
IVF	2554 (69.9%)	308 (73.5%)	0.127 ^b^
ICSI	1099 (30.1%)	111 (26.5%)	
Antral follicle count	11 (7–17)	11 (8–15)	0.977 ^a^
basal FSH	5.32 (4.51–6.50)	5.33 (4.54–6.36)	0.945 ^a^
Stimulation protocol			
Long GnRH agonist protocol	2549 (69.8%)	339 (80.9%)	<0.001 ^b,^*
GnRH antagonist protocol	1104 (30.2%)	80 (19.1%)	
Duration of stimulation (days)	10 (9–12)	11 (10–12)	<0.001 ^a,^*
Total dose of gonadotropins (IU)	1950 (1425–2700)	2350 (1762.5–3000)	<0.001 ^a,^*
LH level on day of HCG administration (mIU/mL)	0.97 (0.65–1.48)	0.66 (0.48–1.25)	<0.001 ^a,^*
Estrogen level on day of HCG administration	3147 (1528.5–4936)	4209 (2936–5000)	<0.001 ^a,^*
Progesterone level on day of HCG administration (ng/mL)	0.7 (0.5–0.9)	1.7 (1.5–2.0)	<0.001 ^a,^*
The endometrial thickness on the HCG trigger day	11 (9.25–13)	11 (9–13)	0.094
Number of oocytes retrieved	15 (7–24)	18 (12–24)	<0.001 a,*
Number of oocytes fertilized	9 (4–15)	11 (8–16)	<0.001 ^a,^*
Number of viable embryos	5 (2–8)	6 (4–9)	< 0.001 ^a,^*
Number of high-quality embryos	4 (1–7)	5 (2–7)	<0.001 ^a,^*
Cumulative live birth rate	1894 (51.8%)	238 (56.8%)	0.054 ^b^

^a^ Two-sample Mann–Whitney test. Values are medians interquartile range. ^b^ Pearson chi-squared test. Values are number(percentage). * Statistical significance. IVF, in vitro fertilization; ICSI, intracytoplasmic sperm injection; LH, luteinizing hormone; GnRH, gonadotropin-releasing hormone; HCG, human chorionic gonadotropin.

**Table 2 jcm-11-07300-t002:** Multivariate logistic regression analysis for the cumulative live birth rate.

	Adjusted ORs	95% CI	*p* Value
Female age (years)	0.936	0.920–0.952	<0.001 *
Body mass index (kg/m^2^)	0.998	0.973–1.023	0.195
Stimulation protocol			
GnRH antagonist protocol **(Ref)**	1	-	
Long GnRH agonist protocol	1.108	0.929–1.321	0.256
Total dose of gonadotropins (IU)	0.998	0.997–0.999	0.006 *
LH level on day of HCG administration (mIU/mL)	0.922	0.873–0.974	0.004 *
Progesterone level on the day of HCG administration	1.109	0.988–1.245	0.080
Number of oocytes retrieved	1.085	1.075–1.095	<0.001 *

CI, confidence interval; OR, odds ratio. GnRH, gonadotropin-releasing hormone; LH, luteinizing hormone; HCG, human chorionic gonadotropin. * Statistical significance.

**Table 3 jcm-11-07300-t003:** Results of multiple regression analysis of embryo quality.

	Number of Viable Embryos	*p* Value	Number of High-Quality Embryos	*p* Value
	β (95% CI)		β (95% CI)	
**Long GnRH agonist protocol**				
Female age	−0.068 (−0.093 to −0.039)	<0.001 *	−0.016 (−0.042 to 0.012)	0.267
Progesterone level on the day of HCG administration (P < 1.5 ng/mL or ≥1.5 ng/mL)	0.026 (−0.01 to 0.706)	0.057	−0.006 (−0.439 to 0.274)	0.650
Number of oocyte retrieved	0.671 (0.281 to 0.305)	<0.001 *	0.673 (0.273 to 0.297)	<0.001 *
**GnRH antagonist protocol**				
Female age	0.016 (−0.036 to 0.028)	0.783	0.011 (−0.041 to 0.025)	0.623
Progesterone level on the day of HCG administration (P < 1.5 ng/mL or ≥1.5 ng/mL)	−0.004 (−0.708 to 0.562)	0.822	−0.006 (−0.763 to 0.547)	0.746
Number of oocyte retrieved	0.807 (0.326 to 0.359)	<0.001 *	0.785 (0.304 to 0.339)	<0.001 *

GnRH, gonadotropin releasing hormone; HCG, human chorionic gonadotropin. * Statistical significance.

## Data Availability

The datasets used and/or analyzed during the current study are available from the corresponding author on reasonable request. It is not possible to share data due to personal data.

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
