# Peer review of "The Impact of Late Follicular Phase Progesterone Elevation on Cumulative Live Birth Rate and Embryo Quality in 4072 Freeze-All Cycles"

_jcm, 2022, doi:10.3390/jcm11247300_

Round 1
Reviewer 1 Report (Previous Reviewer 3)
The authors have successfully addressed the issue raised from my revision.
Author Response
Thank you very much for the comments of the reviewer.
Reviewer 2 Report (New Reviewer)
The manuscript entitled “The impact of late follicular phase progesterone elevation on cumulative live birth rate and embryo quality 4072 freeze-all cycles”. The aim of this study is to investigate the association between P level on trigger day and cumulative live birth rate. The secondary aim of this study is to assess the influence of P level on the trigger day on embryo quality. The manuscript is well written and the contents are informative. However, I would give some concerns for the authors as follows:
1. Some previous studies had demonstrated a deleterious impact of LFEP on oocyte quality. The study should demonstrate the number of MII oocytes between the two groups to see whether significant difference or not in table 1 and put it for logistic regression analysis for the cumulative live birth rate.
2. How many embryos were transferred in the average between the two groups? The number of transferred embryos is strongly associated with the cumulative live birth rate. The study should compare it between the two groups and put it for logistic regression analysis for the cumulative live birth rate.
3. What is the endometrial thickness in the cycles of frozen embryo transfers between the two groups? This factor may be significantly associated with the cumulative live birth rate. This study should compare it between the groups.
4. A study flowchart should be presented in the study.
Author Response
Point 1. Some previous studies had demonstrated a deleterious impact of LFEP on oocyte quality. The study should demonstrate the number of MII oocytes between the two groups to see whether significant difference or not in table 1 and put it for logistic regression analysis for the cumulative live birth rate.
Response: Thank you very much for the comments and suggestions of the reviewers. In the study, our patient population consisted of women who underwent their first IVF or ICSI cycles. In our reproductive centre, the number of MII oocytes in the patients who underwent the IVF were not evaluated and reported. As this is a retrospective study, we can not provide the information of the number of MII oocytes. The retrospective design is the limitation of the study and we have demonstrated it in the discussion part of the study(line 272 to 273).
Point 2. How many embryos were transferred in the average between the two groups? The number of transferred embryos is strongly associated with the cumulative live birth rate. The study should compare it between the two groups and put it for logistic regression analysis for the cumulative live birth rate.
Response: Thank you for these observations. In our reproductive centre, single blastocyst was transferred according to the score from best to worst. Two cleavage embryos were transferred finally when all blastocyts had been transferred without a live birth. The principle of the number of transferred embryos in the groups with progesterone level < 1.5 ng/mL and ≥ 1.5 ng/mL was consistent. It may reduce the bias caused by the number of transferred embryos. We have clarified the principle of the number of transferred embryos in the materials and methods (line 121 to 123 ).
Point 3. What is the endometrial thickness in the cycles of frozen embryo transfers between the two groups? This factor may be significantly associated with the cumulative live birth rate. This study should compare it between the groups.
Response:We appreciate the comments and suggestions of the reviewer. We determined cumulative live birth rate by considering the number of first live births generated during the IVF/ICSI cycles as the numerator and the total number of ovarian stimulation cycles as the denominator. Some patients may undergo more than one cycles of frozen embryo transfer to obtain the first live births. Thus, we can not put the endometrial thickness in the cycles of frozen embryo transfers for logistic regression analysis for the cumulative live birth rate. We added the comparison of endometrial thickness on the HCG trigger day between the two groups in the table 1. As the univariate logistic regression analysis did not show a significant association between cumulative live birth rate and endometrial thickness on the HCG trigger day, we did not put it for multivariate logistic regression.
Point 4. A study flowchart should be presented in the study.
Response:We appreciate the suggestions of the reviewer. We have added the study flowchart as the Figure 1 according to the suggestions of the reviewer.
This manuscript is a resubmission of an earlier submission. The following is a list of the peer review reports and author responses from that submission.
Round 1
Reviewer 1 Report
The subject of this article is the impact of progesterone elevation on cumulative LBR in freeza-all cycle.
This is a very interesting and academically important, unidentified issue.
It included relatively a large number of patients. However, there were some weak points such as retrospective design and not for elective freeze-all cycle.
You analyzed the data at the 1.5ng/mL of progesteronone serum level. But there are many other suggestions about the cut-off level of progesterone for deciding freeze-all strategy with no fresh ET.
It is considered hasty to conclude that progesterone levels don not affect cumulative pregnancy rates through an analysis based on one point of cut-off point.
The study design of this study (retrospective), you can analyze the data with various progesterone levels.
For example, Vuoong LN et al. showed low LBR in fresh ET at the 1.17 or 1.22 ng/mL of P level in their RCT comparing with freeze-all starategy.
Also, Wang A et al., Fertil Steril 2017 suggested the 1.0ng/mL.
Also, there were statistically significant differences at the number of oocytes retrieved, fertilized, viable embryos and high-quality embryos. But the cumulative live birth rate showed no statistical difference. You have to explain these points at discussion section.,
What is the freeze-all indication of 3653 patients who showed lower progesterone level?
Reviewer 2 Report
none
Reviewer 3 Report
The current study investigated the putative effect of late follicular phase progesterone elevation on endometrial receptivity and embryo quality in freeze all cycles. The authors observed no negative impact on cumulative live birth rates and emryo quality. Overall, the manuscript is clear and well written.
Although the study was conducted to address an unsolved question (the effect on embryo quality) I think that the novelty of the study is still limited. In addition some issues should be addressed before considering for publication.
1. As briefly mentioned in the discussion, the freeze-all group is composed by a mix of endometrial preparations. I suggest to consider natural cycle and artificial preparation separately to assess cLBR correctly
2. Why did the authors decide to transfer D3 top quality embryos first and to extend culture to day 5 for the remaining embryos? This represent a crucial bias and cLBR it surely affected by the embryo stage. I suggest to clarify the number of day 3/day 5 ET in each group
3. Please add hormonal and US markers of ovarian reserve (FSH, LH, AFC) in the basal description of the patient population